# Constrain Bias Addition to Train Low-Latency Spiking Neural Networks

**DOI:** 10.3390/brainsci13020319

**Published:** 2023-02-13

**Authors:** Ranxi Lin, Benzhe Dai, Yingkai Zhao, Gang Chen, Huaxiang Lu

**Affiliations:** 1Institute of Semiconductors, Chinese Academy of Sciences, Beijing 100083, China; 2University of Chinese Academy of Sciences, Beijing 100089, China; 3Semiconductor Neural Network Intelligent Perception and Computing Technology Beijing Key Laboratory, Beijing 100083, China; 4Collage of Microelectronics, University of Chinese Academy of Sciences, Beijing 100049, China; 5Materials and Optoelectronics Research Center, University of Chinese Academy of Sciences, Beijing 200031, China

**Keywords:** spiking neural network, backpropagation, Sigma-Delta ADC, neural encoding, pattern recognition

## Abstract

In recent years, a third-generation neural network, namely, spiking neural network, has received plethora of attention in the broad areas of Machine learning and Artificial Intelligence. In this paper, a novel differential-based encoding method is proposed and new spike-based learning rules for backpropagation is derived by constraining the addition of bias voltage in spiking neurons. The proposed differential encoding method can effectively exploit the correlation between the data and improve the performance of the proposed model, and the new learning rule can take complete advantage of the modulation properties of bias on the spike firing threshold. We experiment with the proposed model on the environmental sound dataset RWCP and the image dataset MNIST and Fashion-MNIST, respectively, and assign various conditions to test the learning ability and robustness of the proposed model. The experimental results demonstrate that the proposed model achieves near-optimal results with a smaller time step by maintaining the highest accuracy and robustness with less training data. Among them, in MNIST dataset, compared with the original spiking neural network with the same network structure, we achieved a 0.39% accuracy improvement.

## 1. Introduction

Due to the inherent properties of energy efficiency and biological plausibility, spiking neural networks (SNNs) have made significant advancements lately. Unlike basic artificial neural network (ANN), SNNs make an effort to mimic biological neural networks more closely through discrete events. The SNNs can encode information into a sequence of spikes, and are useful tools for complex spatio-temporal information processing when compared to traditional ANN that use floating-point numbers for operations ([1]). Additionally, the generation of the spike sequence is an asynchronous process, and the operation of the spike is a binary operation, which offers tremendous adaptability to hardware circuits ([2]). Many research institutions have developed neural computing devices based on SNNs, such as SpiNNaker ([3]), TrueNorth ([4]), and Loihi ([5]), among others. The emergence of these platforms has also expanded the potential applications and prolific research prospects of SNNs.

SNNs and ANN differ primarily in two respects. (1) As shown in Figure 1, While the data is infiltrated in to the network, the SNNs must first encode it into a spike train; (2) The spike-firing process of spiking neurons is non-differentiable, which has a negative impact on training. In the field of neuroscience, neural information coding is concerned with the relationship between input signals and the response of individual or group neurons ([6,7,8]). The SNNs primarily utilize two coding methods for spike coding: temporal coding ([9]) and rate coding ([10]). Latency coding ([11]) and phase coding ([12]) are two variations of temporal coding. In a given time window, only one spike is encoded. The input data is included in the precise time of the spiked release, and its value is negatively correlated with the encoding time. At the moment, rate coding is the most predominant coding method used by researchers all over the world. At most, one spike is set to be emitted at each time step within a given time window, and the frequency of spike firing increases as the stimulation intensity increases, resulting in a spike sequence. Rate coding generally requires a sufficient number of time steps to improve coding accuracy, which increases network power consumption. Many researchers in the past have utilized Poisson coding or Bernoulli coding to implement rate coding, which is widely deployed in various deep SNNs. The rate coding method, however, employs statistical models that require sufficient time steps to guarantee coding accuracy. Additionally, it ignores data correlation and presumes that the input data are independent of one another. In recent years, some researchers have used SNNs to directly input real-valued data and treat the first few layers of spiking neural layers as the encoding layer ([13]). This scheme directly increases the scale of the network’s architecture and thereby upsurging the training time and energy consumption.

The post-synaptic neuron accumulates the spike from the preceding layer in the membrane potential of the neuron until it exceeds the threshold, whereupon the post-synaptic neuron emits a spike and reset its membrane potential. However, the spike-firing process is not differentiable, which complicates the backpropagation training of SNNs. At present, the training rules of SNN can be mainly divided into three categories: (1). Learning rules based on biological characteristics. The most typical methods in practice are Spike-Timing- Dependent-Plasticity (STDP) ([14]) and Tempotron ([15]). STDP is a prominent unsupervised learning rule, where the update in synaptic weights is only related to pre-and post-synaptic neuron activity. Tempotron is a gradient descent learning algrithom based on biological characteristics that is suitable for temporal coding, when a classification error occurs, it updates the synaptic weight with temporal information. (2). Surrogate gradient method ([16]). In order to alleviate the dilemma that the SNN is difficult to directly use Backpropagation (BP) for training, the researchers utilize Heaviside function in the process of forward propagation and use a surrogate gradient to approximate the gradient in the backpropagation. For example, in [17], the researchers proposed the spatio-temporal Backpropagation (STBP) algorithm, which utilizes surrogate gradients while computing the gradient of spiking neurons, and further discussed the impact of different surrogate gradients with hyperparameter settings on network performance. Although the surrogate gradient temporarily alleviates the supervised learning training problem of SNNs, there are many choices, and it is difficult to choose the most suitable one. (3). ANN-SNN conversion. The method is divided into two steps: train an ANN and then convert it into an SNN. Ref. [18] first proposed a method to convert ANN to SNNs, then [19] proposed a data-based regularization method, and then further improved the conversion method in [20] to reduce the conversion error; in [21], the researchers converted the residual neural network, which improved the scale of SNNs and in [22], the researchers converted the Tiny YOLO network, making SNNs more widely used in the field of object detection. The conversion-based method can quickly apply SNNs to different tasks and avoid the difficulty of directly training SNNs. However, there will be a loss of accuracy in the conversion process, and the final SNNs often have higher requirements on the encoding time step, which increases the inference latency and the energy consumption of the SNNs in disguised form.

In this paper, we propose a novel encoding scheme along with a spike-based learning rule. The proposed method elucidates a Sigma-Delta encoding method that effectively exploits the correlation between input data and achieves higher encoding gains; we reconsider the role of bias voltage in SNNs and modify the addition mechanism. We deploy the proposed SNNs model to an environmental sound classification task, and image classification tasks, and the results show that our models can achieve competitive results with fewer spikes.

The rest of this article is organized as follows: Section 2 introduces the Sigma-Delta encoding method, and explores the proposed bias addition rule and the new gradient approximation, thereby deriving the backpropagation formula. Section 3 verifies the effectiveness of the proposed encoding method and learning rules through experiments on different tasks and compares them with other models. In Section 4, we summarize and discuss the effectiveness of the outcome obtained in the proposed work.

## 2. Materials and Methods

### 2.1. Spiking Neuron Model

As the most commonly used spiking neuron, the Leaky Integrate-and-Fire neuron (LIF) adopts event-driven simulation strategy and has limited neural computing characteristics ([23]). The first-order differential form of its kinetic equation is as follows:(1)$$\tau \frac{dV(t)}{dt}=-\left(V\left(t\right)-V_{rest}\right)+X\left(t\right)$$
where V(t) represents the membrane potential at the *t*-th moment; $V_{rest}$ represents the resting potential; X(t) is the input at time *t*, including the output spikes of the previous layer and bias voltage; τ is the decay time constant. When the membrane potential V(t) does not exceed the threshold $V_{th}$, the membrane potential decays, and the magnitude is controlled by τ, otherwise, the neuron will fire a spike and reset the membrane potential. We express the firing process with the following formula:(2)$$N\left(t\right)=\left\{\begin{array}{cc} 1, & \mathrm{if}\;V\left(t\right)>V_{th}\\ 0, & \mathrm{if}\;V\left(t\right)\leq V_{th} \end{array}\right.$$ At the *t*-th time step, the value of the membrane potential of the j-th neuron in the l-th layer is given by:(3)$$V_{j}^{l}\left(t\right)=\left(1-\frac{1}{\tau }\right)V_{j}^{l}\left(t-1\right)+\sum_{i=1}^{M_{l-1}}W_{ij}N_{i}^{l-1}\left(t\right)+X_{bias,j}^{l}$$
where $M_{l-1}$ means the number of spiking neurons in l−1-th layer, $X_{bias,j}^{l}$ represents the bias voltage. For the spike reset phase, we use a subtraction-based reset mechanism:(4)$$V_{j}^{l}\left(t\right)=V_{j}^{l}\left(t\right)-V_{th}*N_{j}^{l}\left(t\right)$$ When compared with the method of directly resetting the membrane potential to a fixed value, the subtraction-based mechanism is better suited to training deep SNNs and makes more biological sense ([20,24]).

### 2.2. Spike Encoding Method

Most of the present SNNs encoding methods such as time coding and rate coding involves the encoding of input numerical data. This characteristic assumes that the input data are judged to be independent of one another. But there are correlations between data at different points in a single image or spectrogram of a single speech signal, especially in spectrograms, where each row represents a subband. Differential Pulse Coding Method (DPCM) is a common coding method in classical speech coding schemes that can eliminate information redundancy in the spoken signal and achieve commendable data compression. DPCM does not encode the input data directly, but rather make use of the difference between adjacent sampled signals. This technology has been applied not just in the field of communication, but also in digital picture processing ([25,26]). However, the DPCM technique has to rebuild the sampled value at the previous time in the calculation process and then calculate the difference value with the current input, which increases the computing complexity of the encoding process.

The Sigma-Delta ADC is a very common circuit structure in the field of integrated circuits, which can convert analog signals to digital signals. We model the process by which a first-order Sigma-Delta ADC outputs a digital signal. In some previous studies, researchers carried out the process of combining Sigma-Delta ADC with SNNs. It is used as the spiking neurons to process spike trains in [27,28]; in [29], the researchers add it to each layer of the network and propose an STDP-like learning mechanism; researchers in [30] quantify the activation of ANN and convert them to SNNs using a method similar to Sigma-Delta modulation. We utilize Sigma-Delta ADC as the rate encoder and simplify the model to realize better computational efficiency. The Sigma-Delta ADC encoding process, shown in Figure 2, is similar to DPCM, but employs fixed return values instead of predictors. When the input is a picture or a spectrogram that involves a numeric matrix, we set the data in row i and column j as x(i,j) and encode the input data along the rows. It is important to note that the number of rows of input data is equal to the number of encoders. The encoding process of i-th row is as follows:(5)$$e\left(i,j\right)=x\left(i,j\right)-\hat{x}\left(i,j-1\right)$$
(6)I(i,j)=e(i,j)+I(i,j−1)
(7)$$\left\{\begin{array}{c} y\left(i,j\right)=1,\hat{x}\left(i,j\right)=1,\mathrm{if}\;I\left(i,j\right)>V_{th}\\ y\left(i,j\right)=0,\hat{x}\left(i,j\right)=-1,\mathrm{if}\;I\left(i,j\right)\leq V_{th} \end{array}\right.$$
where e(i,j) represents the prediction error, I(i,j) denotes the value of the integrator, $\hat{x}\left(i,j\right)$ represents the predicted value, and is determined by the Formula (Equation 7), y(i,j) is the output of the encoder, and $V_{th}$ is the benchmark of the comparator. When the time window is N, the above formula will loop N times, but at the end of each loop, the value of the integrator will be retained and used for the next loop.

It can be seen from the Formulas (Equation 5)–(Equation 7) that any data will be affected by the already encoded data during the encoding process. If the input data is a spectrogram, the data of any subband will be disturbed by the data of the same subband during the encoding process. For image data, we still input along the rows during the experiment. Note that the input range for Sigma-Delta encoder(SDE) is [−1, 1]. We use linear and nonlinear methods to map the input data to this range.The mapping formula is as follows:(8)$$Linear:x=\frac{X\times 2}{X_{max}-X_{min}}-\frac{X_{max}+X_{min}}{X_{max}-X_{min}}$$
(9)$$Sin:x=\frac{X\times \pi }{X_{max}-X_{min}}-\frac{\pi (X_{max}+X_{min})}{2(X_{max}-X_{min})}$$
where $X_{max}$ and $X_{min}$ represent the upper and lower bounds of the input value, respectively. Linear mapping will not affect the distribution of the data, while sinusoidal mapping will increase the difference between higher and lower value regions.

### 2.3. Supervised Training of Deep Spiking Neural Network

#### 2.3.1. Spiking Neuron Gradient Estimation

ANN often uses the ReLU function as an activation function. The input and output characteristics are shown in Figure 3a. If and only when the input value exceeds zero, the information will continue to be transmitted; otherwise, the output is zero. The ReLU function introduces nonlinearity and alleviates the problem of exploding or vanishing gradients when training with the gradient backpropagation algorithm ([31]). As shown in Figure 3b, we can observe that there is also a positive correlation between the spike output frequency and the inputs of Integrate-and-Fire (IF) neurons in SNNs. But, as their outputs are spike trains rather than continuous values, the spike firing process is non-differentiable, which makes SNNs difficult to train directly by using the backpropagation algorithm.

In this study, the role of bias voltage in spiking neurons is reconsidered. By developing a new bias addition mechanism, a novel gradient approximation method is proposed based on the new rule’s input and output characteristics. In this work, it is observed that the gradient of a spiking neuron should be divided into two parts based on the range of the input: when the input is less than or equal to zero, the gradient is considered to be zero, and when the input is greater than zero, the gradient is calculated, and the detailed derivation of the gradient is given in the following sentences.

Spiking neurons in SNNs fire only when the membrane potential exceeds the threshold, the source of the membrane potential can be separated into three parts, as shown in Figure 4: the membrane potential at the last time-step, the input, and the bias. The bias voltage value and the spike firing rate have a positive association. Due to the fact that only the membrane potential and the input change during the network inference process, the neuron’s actual spike firing threshold ${\hat{V}}_{th}$ is equal to $V_{th}-X_{bias}$. However, in the previous research, the processing of bias current is mainly divided into two cases: (1). The bias is directly set to 0 and not as a variable during the network learning process; (2). Like ANN, the bias is added at each moment and updated together with the weights during the network learning process. In case 1, since the bias is permanently forced to zero, it is equivalent to ignoring the regulation effect of bias on the threshold; while in case 2, adding bias voltage at every time-step undoubtedly violates the event-driven characteristics of SNNs because When time T→∞, in the absence of any input, the spiking neuron will spontaneously emit spikes, which has a negative impact on training and reasoning.

For the above two cases, we modify the bias addition mechanism. We want to think of the bias as the reference voltage in the spiking neuron, which means that in the absence of any input, the membrane potential of the spiking neuron will always be $X_{bias}$. When the output spikes of the previous layer arrive, for the change of the membrane potential of the IF neuron, we describe it with the following formula:(10)$$\left\{\begin{array}{c} V_{j}^{l}\left(t_{-}\right)=V_{j}^{l}\left(t-1\right)+\sum_{i=1}^{M_{l-1}}W_{ij}N_{i}^{l-1}\left(t\right)\\ V_{j}^{l}\left(t\right)=V_{j}^{l}\left(t_{-}\right)+\left(V_{th}-X_{bias,j}^{l}\right)*N_{j}^{l}\left(t\right) \end{array}\right.$$
where $V_{j}^{l}\left(t_{-}\right)$ denotes the membrane potential of the previous layer when the spike just arrived, $V_{j}^{l}\left(t\right)$ presents the membrane potential after the spike is emitted. When the spiking neuron emits a spike, the membrane potential is reset according to the Formula (Equation 4). We set the bias to be consumed during the reset process, and then the bias will be added at the end of the moment, the total process is shown in the right side of Figure 4. At the same time, we set the final membrane potential to be $V_{j}^{l}\left(T\right)$ after the inference process is over. In the whole process, a single neuron satisfies:(11)$$\sum_{t=1}^{T}\sum_{i=1}^{M_{l-1}}W_{ij}N_{i}^{l-1}\left(t\right)+\left(N_{j}^{l}+1\right)X_{bias,j}^{l}=N_{j}^{l}V_{th}+V_{j}^{l}\left(T\right)$$
where $N_{j}^{l}$ is the total number of transmitted spikes. In order to reduce the difficulty of solving, we ignore the final membrane potential $V_{j}^{l}\left(T\right)$ because $V_{j}^{l}\left(T\right)$ must contain bias, so a bias needs to be subtracted from the left side of the equation, and the final Equation (Equation 11) evolves into the following form:(12)$$\sum_{t=1}^{T}\sum_{i=1}^{M_{l-1}}W_{ij}N_{i}^{l-1}\left(t\right)+N_{j}^{l}X_{bias,j}^{l}=N_{j}^{l}V_{th}$$ For a single spike, its value is generally set to 1, so it can be seen that $N_{j}^{l}$ is the final output of the neuron. Let $X_{j}^{l}=\sum_{t=1}^{T}\sum_{i=1}^{M_{l-1}}W_{ij}N_{i}^{l-1}\left(t\right)$ as the total input from the previous layer, we can get:(13)$$N_{j}^{l}=\frac{X_{j}^{l}}{V_{th}-X_{bias,j}^{l}}$$ Since then, we have approximated the input-output relationship of the IF neuron, and can be approximately applied at each time step. The process is similar to the LIF neuron, but in the process of calculating the membrane potential, the decay factor needs to be considered like the Formula (Equation 3), then there is more loss when approximating the Formula (Equation 13). Therefore, only IF neurons are used in all experiments in this paper.

#### 2.3.2. Spike-Based Backpropagation Algorithm

In Formula (Equation 13), we obtain the input-output relationship of the spiking neuron. By simply derivation, we can easily obtain the following formula:(14)$$\begin{array}{cc} \frac{dN_{j}^{l}}{dX_{j}^{l}} & =\frac{1}{V_{th}-X_{bias,j}^{l}} \end{array}$$(15)$$\begin{array}{cc} \frac{dN_{j}^{l}}{dX_{bias,j}^{l}} & =\frac{N_{j}^{l}}{V_{th}-X_{bias,j}^{l}} \end{array}$$ After obtaining the approximate gradient of the neuron, we design our backpropagation gradient algorithm using the BackPropagation Through Time (BPTT) concept. We define the output layer of the network as a linear layer that emits no spikes and accumulates the membrane potential to determine the category based on the magnitude of the membrane potential. For a M classification problem, we utilize Mean-Square-Error (MSE) as our loss function with:(16)$$E=\frac{1}{2}\sum_{m=1}^{M}{\left({\hat{y}}_{m}-y_{m}\right)}^{2}$$
where $y_{m}=\frac{V_{m}^{L}\left(T\right)}{TV_{th}}$, during backpropagation, we use the chain rule. We set the L-th layer as the output layer, and then for the output layer neurons, the gradient calculation is as follows:(17)$$\begin{array}{cc} \frac{dE}{dX_{m}^{L}} & =\frac{dE}{dy_{m}}\frac{dy_{m}}{dX_{m}^{L}} \end{array}$$(18)$$\begin{array}{cc} \; & =-\frac{\left({\hat{y}}_{m}-y_{m}\right)}{TV_{th}} \end{array}$$(19)$$\begin{array}{cc} \frac{dE}{dX_{bias,m}^{L}} & =\frac{dE}{dy_{m}}\frac{dy_{m}}{dX_{bias,m}^{L}} \end{array}$$(20)$$\begin{array}{cc} \; & =-\frac{\left({\hat{y}}_{m}-y_{m}\right)}{V_{th}} \end{array}$$ For the hidden layer, to the j-th neuron in the l-th layer, the gradient calculation formula at the moment $t_{0}$ is as follows:(21)$$\begin{array}{cc} \frac{dE}{dN_{j}^{l,t_{0}}} & =\sum_{m=1}^{M}\frac{dE}{dy_{m}^{T}}\left(\frac{dy_{m}^{T}}{dy_{m}^{t_{0}}}\frac{dy_{m}^{t_{0}}}{dN_{j}^{l,t_{0}}}+\sum_{t=t_{0}+1}^{T}\frac{dy_{m}^{T}}{dN_{j}^{l,t}}\frac{dN_{j}^{l,t}}{dN_{j}^{l,t_{0}}}\right) \end{array}$$(22)$$\begin{array}{cc} \frac{dy_{m}^{t_{0}}}{dN_{j}^{l,t_{0}}} & =\sum_{k=1}^{N^{l+1}}\frac{dy_{m}^{t_{0}}}{dN_{k}^{l+1,t_{0}}}\frac{dN_{k}^{l+1,t_{0}}}{dX_{k}^{l+1,t_{0}}}\frac{dX_{k}^{l+1,t_{0}}}{dN_{j}^{l,t_{0}}} \end{array}$$(23)$$\begin{array}{cc} \frac{dN_{j}^{l,t}}{dN_{j}^{l,t_{0}}} & =\frac{dN_{j}^{l,t}}{dV_{j}^{l}\left(t\right)}\frac{dV_{j}^{l}\left(t\right)}{dV_{j}^{l}\left(t_{0}\right)}\frac{dV_{j}^{l}\left(t_{0}\right)}{dN_{j}^{l,t_{0}}} \end{array}$$

According to the Formula (Equation 10), the first two items of the Formula (23) can be known:(24)$$\begin{array}{cc} \frac{dN_{j}^{l,t}}{dV_{j}^{l}\left(t\right)} & =\frac{dN_{j}^{l,t}}{dV_{j}^{l}\left(t\right)}\frac{dV_{j}^{l}\left(t\right)}{dX_{j}^{l,t}}=\frac{N_{j}^{l,t}}{V_{th}-X_{bias,j}^{l}} \end{array}$$(25)$$\begin{array}{cc} \frac{dV_{j}^{l}\left(t\right)}{dV_{j}^{l}\left(t_{0}\right)} & =1 \end{array}$$ Considering the case of $N_{j}^{l,t_{0}}=0$, the last term of Formula (23) cannot be calculated directly, but can be eliminated by the chain rule. Finally, we get the gradient of the loss function to the weight $W^{l}$ and bias $X_{bias,j}^{l}$ respectively:(26)$$\begin{array}{cc} \frac{dE}{dW^{l}} & =\sum_{t=1}^{T}\sum_{k=1}^{N^{l+1}}\frac{dE}{dN_{j}^{l+1,t}}\frac{dN_{j}^{l+1,t}}{dW^{l}} \end{array}$$(27)$$\begin{array}{cc} \frac{dE}{dX_{bias,j}^{l}} & =\sum_{t=1}^{T}\sum_{k=1}^{N^{l+1}}\frac{dE}{dN_{j}^{l,t}}\frac{dN_{j}^{l,t}}{dX_{bias,j}^{l}} \end{array}$$

The pseudocode for the entire training phase is shown below:
**Algorithm 1:** Training process of the proposed SNN model and SDE method.**Input:** Network input $X_{i,j}$, i∈[0,M], j∈[0,N]; sample label *Y*; time windows *T*; 1:**initialization** parameters and states of network $\omega_{ij}^{n}$, $bias_{i}^{n}$, n=1,2,⋯N; states of the integrator in SDE I(i,j);2:**Forward**3:**for** t∈[0,T]**do**4:    **for** j∈[0,N] **do**5:        # Each SDE encodes an entire line of input data6:        $Spike_{(}i,j)=SDE\left(X_{i,j}\right)$7:    **end for**8:    **for** n∈[0,N] **do**9:        $Spike^{n}\leftarrow \omega \cdot Spike^{n-1}$10:    **end for**11:**end for**12:**Backforward**13:Calculate loss function $L=MSE(Spike^{N-1},Y)$14:Calculate gradient through BPTT: $\Delta_{\omega }\leftarrow {▿}_{\omega }L{,}_{bias}\leftarrow {▿}_{bias}L$15:Update Parameters16:**Reset**17:Reset the membrane V(T−1) of the spiking neuron and the state *I* of the SDE integrator to 0.


## 3. Results

### 3.1. Experimental Setup

#### 3.1.1. Experimental Environment

To validate the effectiveness of the proposed encoding method and learning rule, we apply the proposed model to environmental sound classification tasks and image classification tasks. The network structures and hyperparameters we use are different for different tasks, The training process of the network is shown in Algorithm 1. For different tasks, the input data needs to be preprocessed differently.. It is worth noting that the input of the network needs to be input to the network at multiple time steps in SNNs. Hence, when using the dropout layer, it is a necessary requirement that the mask is the same for the same set of data input at each time step, which is mentioned in [32]. Our model is constructed using the PyTorch-based SpikingJelly framework, which facilitates faster inference with GPU acceleration ([33]). The environmental sound classification experiment is run on Intel(R) Core(TM) i7-8700 CPU, and the image classification task is performed on NVIDIA TitanXP Graphics card.

#### 3.1.2. Datasets

In the experimental stage, the data set used for the environmental sound recognition task is the Real World Computing Partnership (RWCP) sound scene dataset dataset ([34]). Following the settings of the previous research, we selected ten categories: bells, bottle, buzzer, cymbals, horn, kara, metal, phone, ring, and whistle, with durations ranging from 0.5 s to 3.0 s. We randomly select 80 files from each class, picking half for training and the other half for testing. In order to test the effectiveness of learning, we add noise to the test set while the training set does not change. Under such mismatched conditions, the test set with noises of different signal-to-noise ratios is tested. The experimental results in this part are all the average values of ten groups of experimental results. We use the MNIST dataset and the Fashion-MNIST dataset for image classification tasks. The MNIST dataset is one of the most classic datasets in deep learning, which contains ten different handwritten digits and the input size is 28×28 ([35]). It contains ten classes, and the training set has a total of 60,000 samples, each class contains 6000 samples. The test set contains 10,000 samples, 1000 samples per class. The Fashion-MNIST and MNIST dataset have the same number of sample classes and samples. Its data is more complex, in which the content of the pictures includes various clothes, the input size is 28×28, and the number of training and test sets is the same as in MNIST ([36]).

#### 3.1.3. Experimental Chapter Arrangement

Our experiment is divided into three parts. In Section 3.2, we completed the task of environmental sound classification in the RWCP dataset. We briefly introduced the feature extraction algorithm we used and the whole process of feature extraction and coding. We perform our experiment under different normalized mapping modes and check the network performance under different time Windows. In order to verify the network’s robustness, noises with various signal-to-noise ratios were introduced to the test set in addition to clean data sets for training and testing.

In Section 3.3, we used the MNIST and Fashion-MNIST data sets for image classification tasks. We evaluate the network’s performance with various numbers of codes, compare the outcomes with those from other SNN models, and display the results in a table. By lowering the amount of training samples, we further test the network’s capacity for learning in Section 3.4. Finally, We analyzed the algorithm efficiency in Section 3.5.

### 3.2. Environmental Sound Classification

#### 3.2.1. Constant-Q Transform

Constant-Q transformation (CQT) refers to a technique for converting a time-domain signal *x*(*n*) into the time-domain frequency domain, and the center frequencies of the frequency bins are geometrically spaced. The conversion formula from discrete time-domain signal *x*(*n*) to frequency domain *X*(*k*, *n*) is as follows:(28)$$X^{CQ}\left(k,n\right)=\sum_{j=n-\lfloor N_{k}/2\rfloor }^{n+\lfloor N_{k}/2\rfloor }x\left(j\right)a_{k}^{*}\left(j-n+N_{k}/2\right)$$
where *k* is the indexes of the frequency bins and denoted by $N_{bins}$ in Table 1. ⌊·⌋ denotes rounding towards negative infinity, $a_{k}^{*}$ represents the complex conjugate of $a_{k}\left(n\right)$, details can be found in [37]. The calculation formula of the center frequency $f_{k}$ of each frequency bin is:(29)$$f_{k}=f_{1}2^{\frac{k-1}{B}}$$
where $f_{1}$ is the initial center frequency of the lowest frequency bin. The value of *Q* is determined by
(30)$$Q=\frac{q}{\Delta \omega (2^{\frac{1}{B}}-1)}$$
where Δ
ω has different values for different window functions, *q* is a scaling factor and usually set to 1, *B* represents the number of bins per octave which is important to the performance of CQT. If the value of *B* is set very high, then the interval between the center frequencies will be very small, for the same band length, we can obtain more frequency bins, which can be regarded as realizing the oversampling of the frequency axis. In our experiment, in order to limit the size of the spectrogram, we set the value of *B* as 3.

#### 3.2.2. Network Structure and Parameter Setting

We use the CQT to extract audio features from environmental sound data and create spectrograms. The main advantage of CQT is that it devoid of suffering from uniform time-frequency resolution because the bandwidth of its subband to its center frequency ratio is a constant value. Therefore, high frequency waves have a larger bandwidth and more substantial temporal resolution at high frequencies read the path of quick changing overtones; In contrast, its bandwidth is relatively narrow for low-frequency waves, while it has a higher-frequency resolution to resolve related notes. In reality, the CQT can be used to evaluate the audio signal more thoroughly since the energy of the sound signal is primarily focused in the low-frequency range. We use Librosa library to complete the experiment of the feature extraction part ([38]). The hyperparameters we set in the experiments are shown in Table 1.

Due to the different lengths of the environmental sound segments employed, the dimensions of the spectrogram produced by CQT after its extraction also vary. Therefore, we divide the temporal dimension of the spectrogram into T-segments. Each segment has different sampling numbers for sound signals with different lengths. Then, we average each segment to obtain a spectrogram with a defined scale. If M cochlear filters are used and T is the length of the time axis, after segmenting the temporal dimension, we obtain a spectrogram of size (M, T). Therefore, we input the spectrogram directly to the SDE, and each SDE encodes only one data row, so we need M SDEs to perform the encoding process, which is shown in the Figure 5. After the encoding process, we flat the spike matrix into a 1-dimensional vector and input it into the fully connected SNNs. The network structure is 800×50×10, and the threshold of the last layer is set to infinity. In the initialization phase of the network, the weights are initialized according to the normal distribution, the fixed bias is 1 and the threshold is 3. Since the RWCP dataset is relatively small, the default batch size is set to 1.

#### 3.2.3. Classification Performance

We first conduct experiments under the condition of clean signal. As shown in Table 2, for different mapping methods, even the same time window will show differences. When the time window is equal to 20, the network using linear mapping achieves the best performance, and the average result of ten experiments achieves 99.85%. When the time window is 16, the sinusoidal mapping method achieves the best performance of 99.68%. In Table 3, we compare our results with other SNNs models and traditional MFCC-HMM methods; we can see that our proposed approach with a fully connected SNN model achieves results close to the best and even exceeds some SNN models that include convolutional operations. We further test the robustness of the model by adding noise to the test data, while still using clean sound for training and testing the ability of the network to extract audio features by constructing mismatched conditions. It can be seen from Table 4 that under the mismatched condition, the network using sinusoidal mapping can achieve better robustness, the optimal performance is achieved when the time window is 12, and the average precision reaches 71.49%, while the optimal performance of linear scaling is 67.23%, the time window is 14. In this regard, we consider sinusoidal mapping that can introduce more nonlinear transformations, which can improve the robustness of the network to a certain extent. Moreover, we find that the performance of the network first increases and then decreases as the time window increases, which is consistent with our expectations. When the time windows are too small, the encoding of the input feature image is relatively coarse. At this point, increasing the time windows may benefit feature extraction. However, if the time windows are too large, the network will be overfitted, the slight differences in the network can also affect the performance of the network.

### 3.3. Image Classification

#### 3.3.1. Network Structure and Parameter Settings

Compared to environmental sound classification task, most current SNNs tasks choose to demonstrate network performance on image datasets, and the relevant network parameters are shown in Table 5. The network structure we choose is 15C5-40C5-FC300-FC10, where 15C5 refers to the output channel of the first layer, which is 15, and the size of the convolution kernel is 5×5. FC300 means that the number of neurons in the fully connected layer is 300. We set the parameter of the dropout layer to 0.2 and employ only linear mapping when we use SDE. We use stochastic gradient descent to update the network parameters and cosine annealing to control the change in the learning rate. The period of cosine annealing is 100, and the period of the whole training is 200, which means the learning rate at the beginning and end of the training is the maximum learning rate. At the 100th epoch, the learning rate decreases to its lowest value. The advantage of this setting is that the value of the loss function of the network is very small in the later training phase, and it is difficult for a small learning rate to break the optimal local solution. Therefore, using a higher learning rate at the end of network training allows the network to explore a larger space to achieve superior performance.

#### 3.3.2. MNIST Dataset

As one of the most widely used datasets for classification tasks, MNIST has been used by many SNN models trained on spikes to test performance. In the Table 6, we show the performance of our model for different time windows. With a number equal to 12, the performance of the network has reached 99.37%. With the increase of the time window to 14, the classification accuracy reaches the highest of 99.60%. We further compare the results of our network with other models with similar structures and present the results in the Table 7. For other networks with similar structures, our network shows advantages in both the time windows and the network performance.

To examine the effect of different time windows on the performance of the classification task, we plot the training curves in the Figure 6. From the different training curves in the Figure 6a, we can see that our network can achieve more than 97% accuracy in the initial stage of training, and as the training time increases, the accuracy of the network continues to increase. In the later stages of training, the performance of the network is still very stable, even though the learning rate continues to increase at this time. A lower time window can better highlight the advantages of the SNN’s low energy consumption and reduce the time consumption for inference. We replaced the coding method with Poisson coding to test the influence of SDE on model performance. The training curve is shown in Figure 6b. It can be observed that the model based on SDE has better performance and faster training accuracy. Since the Sigma-Delta ADC is full-fledged method in the field of analog circuits and the SDE is equivalent to a first-order Sigma-Delta ADC, it is very easy to implement in hardware.

#### 3.3.3. Fashion-MNIST Dataset

For the Fashion-MNIST dataset, the image content is more complex and challenging than the MNIST dataset. In the Table 8, we show the performance of other SNNs models on this dataset and show the performance of our model under a different number of codes. When the time window is 20, the model accuracy reaches 90.14%, and when the time window is increased to 100, the network performance reaches 91.71%. Our model can achieve high accuracy with a small time window, which also reflects the low power consumption of SNNs.

### 3.4. Training with Less Data

To test the learning ability of the network, we control the amount of data used to train the model and test with the three datasets above. For the RWCP dataset, We adjusted the number of training data. For the MNIST and Fashion-MNIST datasets, we set the ratio of the training data to the training set to 0.1 and 0.5, respectively, and the time windows are 14 and 20 with respect to the MNIST and Fashion-MNIST datasets.

The experimental results for the RWCP dataset are shown in Table 9. When we reduce the ratio of the training data to 0.1, our SNNs model can still achieve an accuracy of 98.97% and 97.318%, respectively, and even under mismatched conditions, the average accuracy can reach 70.53%. When we increased the ratio to 0.9, the performance of the network is significantly improved. The average accuracy reached 99.70% and 99.75% without adding noise, and the average performance under mismatched conditions is 68.00% and 73.20%, respectively.

As shown in Figure 7a, for the MNIST dataset, when the number of data points used for training is 6000, our proposed SNNs model achieves 98.41%, and when the training number increases to 30,000, the training accuracy further rises to 99.22%. For the Fashion-MNIST dataset, the training curve is shown in Figure 7b, we use 10% of the training set data to train, the model performance reaches 84.07%, and its training curve shows a trend of overfitting; when using half the number of training sets during training, the network accuracy improves to 87.74%. We believe that the images in the Fashion-MNIST dataset are more complex than those in MNIST. Although the amount of data used for training is the same, Fashion-MNIST’s image feature extraction is more difficult, and the model is more prone to overfitting.

### 3.5. Algorithm Efficiency

We compare the proposed SNN model with SNNs based on native IF neurons. The network model used is still LeNet5, and the dataset is MNIST. We first compare the average running time and memory consumption of each epoch, reflecting the efficiency of the algorithm. The running time of a single epoch for SNNs based on primary IF neurons is 94 s, while the running time of a single epoch for our proposed SNN model is 100 s. This is due to the need to determine whether to add bias when implementing the code. In terms of memory consumption, as can be seen in Figure 8a, our model occupies about 889 MB, while the native IF neuron model consumes 1003 MB, which is related to the spike sparsity of the network. Note that the approximate gradient we derived has an obvious advantage over the surrogate gradient: if the spike neuron emits no spikes at time t, the gradient at time t is 0, which effectively reduces memory consumption.

We further studied the performance of the network. We tested the model under the same time window. The training curves of the two models are shown in the Figure 8b. It can be seen that our model is not only much higher in precision than the model using native IF neurons, but also has obvious advantages in convergence speed.

## 4. Discussion

For SNNs, spike encoding is a crucial process, and the question of how to achieve a higher coding benefits is one worth exploring, the efficiency of encoded spike trains has an important impact on the performance of the model. The proposed SDE can be applied to sound recognition and image classification tasks and uses the correlation between data to effectively reduce the time windows. In traditional deep neural networks, the bias measures the ease with which positive and negative excitations are generated, while in SNNs, the bias adjusts the ease with which spikes are fired. In previous studies, researchers have not emphasized the effect of bias. We believe that the adjustment of the bias is equivalent to changing the spike firing threshold. In this work, we usefully restrict the addition of bias to tight coupling to the spike, which allows us to obtain identities for the input and output of a single neuron throughout the inference process and to obtain gradient approximations. The bias voltage is different for different neurons, and the actual spike firing threshold is also different, reflecting the heterogeneity of biological neurons and consistent with biological systems.

We apply the proposed SNNs model to the classification of environmental sounds. Compared to images, sound signals contain rich spatiotemporal features and are more suitable for evaluating the performance of SNNs. For the test set without noise, our average precision reaches a maximum of 99.85%, and for the test set with noise, our average precision reaches 71.49%, reflecting the robustness of the network. We also test the performance of the proposed SNN model with only a few samples. The experiments show that using only 80 audio samples for training and 720 audio samples for testing, the network achieves the highest average precision of 98.97% without adding noise. The average accuracy when noise is added reaches 70.53%, which reflects the excellent learning ability of the SNNs. We verify the effectiveness of the proposed learning rule and SDE on the MNIST and Fashion-MNIST image datasets. For the MNIST dataset, compared to previous SNN studies with similar network structures, our network achieves near-optimal results in only 14 time steps, and for the Fashion-MNIST dataset, the network performance also achieves 90.26% when the time window is equal to 20, which is better than most networks.

Apart from the conception of novel SNN coding method which is highly viable for hardware implementation, the proposed bias addition mechanism is also consistent with the event-driven nature of SNNs. Compared to most other SNNs models, our network uses fewer time windows, which means that our model spikes more sparsely during inference. This aspect affirms the advantage of low energy consumption of SNNs and reduces the response time of the network, which is beneficial for its application in practical scenarios. During the training process, in quest of speeding up the training, techniques such as gradient normalization and weight constraints are avoided. Hence, suitability facet for on-chip training of neuromorphic chips is assured with great aplomb.

## Figures and Tables

**Figure 1 brainsci-13-00319-f001:**
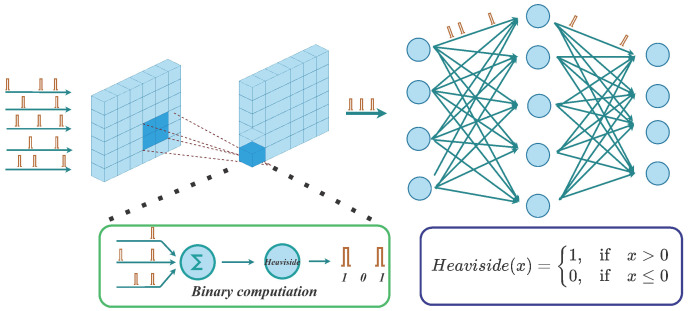
Information transfer and structure diagram of Deep SNNs. If the input type is a floating point, it is usually encoded as a spike train and injected into the network at different time steps. When the membrane potential exceeds a threshold, the spiking neurons can accumulate the spikes from the presynaptic neurons and fire; otherwise, the output is zero.

**Figure 2 brainsci-13-00319-f002:**
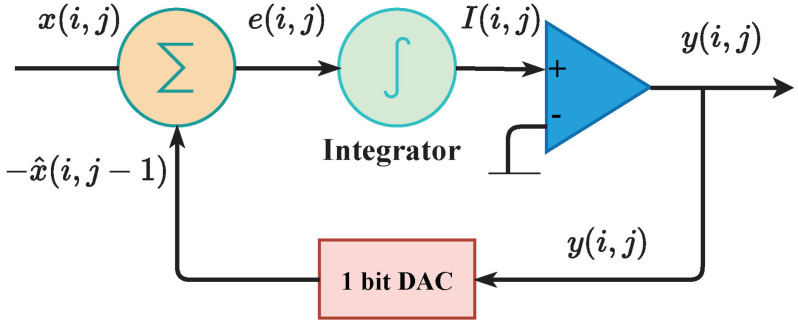
The encoding process of the SDE. Because the integrator’s value I(i,j) is not reset during encoding, each input value is affected by others. The return value of 1-bit DAC $\hat{x}\left(i,j\right)$∈$\left\{-1,1\right\}$.

**Figure 3 brainsci-13-00319-f003:**
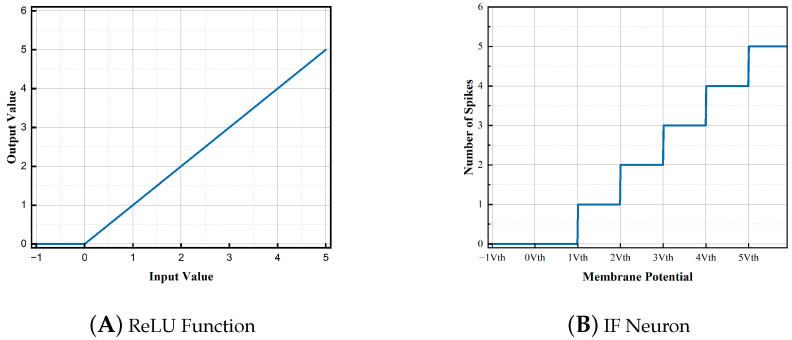
The input-output curve of different neurons. (**A**) When the input is higher than zero, the output is equal to the input, and the gradient is 1; otherwise, the output and gradient are both 0. (**B**) Spiking neurons fire only when the membrane potential exceeds the threshold, and the process is non-differentiable.

**Figure 4 brainsci-13-00319-f004:**
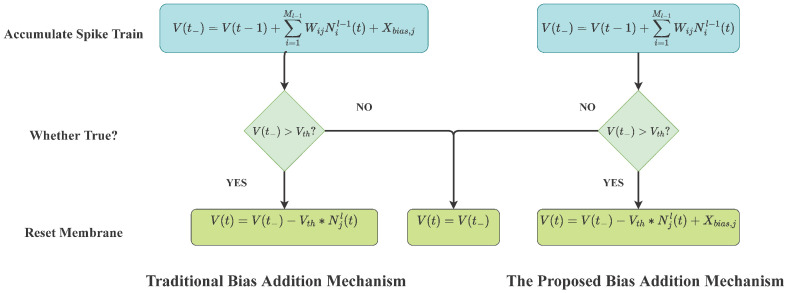
The computational process of IF neurons. The left side is the existing bias addition mechanism, which adds bias in the accumulate spike train phase and triggers a spike when the membrane potential exceeds the threshold; the new bias addition mechanism adds bias only when the membrane potential is reset to ensure that the membrane potential does not increase in the absence of input.

**Figure 5 brainsci-13-00319-f005:**
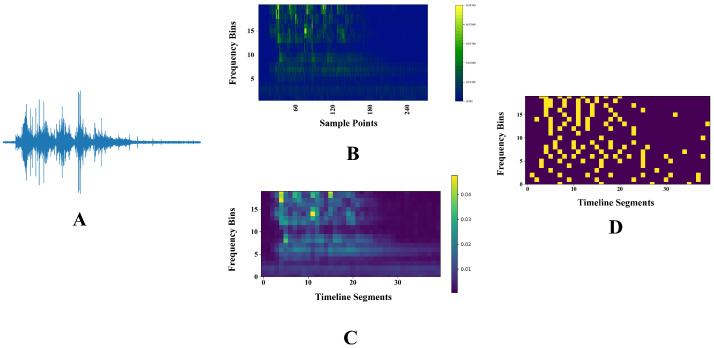
We choose ’horn’ to illustrate the Sigma-Delta audio encoding process. (**A**) The time-domain plot of the horn. (**B**) CQT is employed to obtain the spectrogram. The sampling rate is 48 kHz, and the number of samples between consecutive frames is 128. (**C**) To reduce the scale of the spectrogram and the number of spiking neurons in the first layer of the full-connected SNNs, each spectrogram of the subband is divided into specified timeline segments, with the points in each segment being averaged. (**D**) Encode the scaled spectrogram using SDE, noting that each SDE encodes only one row of data.

**Figure 6 brainsci-13-00319-f006:**
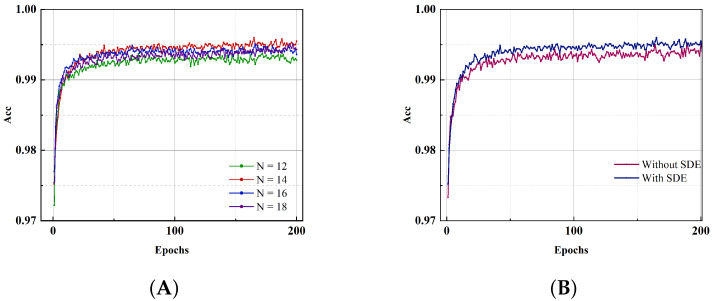
Training curves for different encoding numbers and types. (**A**) The effect of different encoding numbers on network training and performance. (**B**) The effect of different encoding methods.

**Figure 7 brainsci-13-00319-f007:**
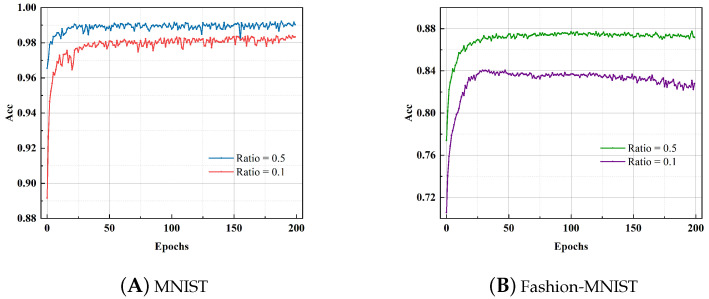
The training curves of the proposed SNNs model for the MNIST dataset and Fashion-MNIST dataset when the ratio of training data to the training set is 0.5 and 0.1, respectively. (**A**). Although there are fewer training data sets for the MNIST dataset, the network training curve is still flat. Even with a ratio of 0.1, the training accuracy still reaches 98.41%. (**B**). The Fashion-MNIST dataset’s image content is comparatively sophisticated. The training curve shows that the network tends to overfit when there are fewer data available for training.

**Figure 8 brainsci-13-00319-f008:**
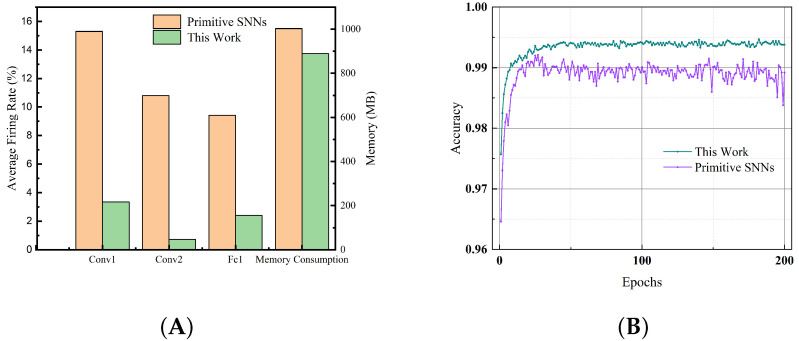
(**A**) The average firing rate of the spiking layer and memory usage per epoch. (**B**) The training curve of our model and the primitive SNNs, the final performance of our model is not only better than the original SNNs, but also converges faster.

**Table 1 brainsci-13-00319-t001:** Parameters set for the RWCP dataset.

Network Parameter	Description	Value
T	Simlulation time	20
$V_{th}$	Threshold of spiking neurons	3.0
b	bias voltage	1.0
$N_{bins}$	The number of filters using in CQT	20
$T_{seg}$	The Number of Timeline segments	40
λ	Learning rate	8 × 10${}^{-3}$

**Table 2 brainsci-13-00319-t002:** Accuracy of different mapping methods under different encoding numbers.

Linear			Sine		
Time Window	Clean	Average	Time Window	Clean	Average
N = 12	99.83%	66.46%	N = 12	99.53%	71.49%
N = 14	99.73%	67.23%	N = 14	99.53%	71.38%
N = 16	99.63%	66.73%	N = 16	99.68%	70.53%
N = 18	99.80%	66.12%	N = 18	99.38%	69.27%
N = 20	99.85%	66.42%	N = 20	99.65%	69.11%

**Table 3 brainsci-13-00319-t003:** Comparison of performance with different models.

Methods	Accuracy	Methods	Accuracy
MFCC-HMM ([39])	99.00%	SPEC-DNN ([40])	100.00%
LSF-SNN ([41])	98.50%	SPEC-CNN ([40])	99.83%
SOM-SNN ([39])	99.60%	Peterson et al. ([42])	99.30%
KP-SNN ([43])	100.00%		
This work	99.85%		

**Table 4 brainsci-13-00319-t004:** Performance evaluation of various models in mismatched conditions.

Methods	Clean	20 dB	10 dB	0 dB	–5 dB	Average
MFCC-HMM ([39])	99.00%	62.10%	34.40%	21.80%	19.50%	47.30%
SPEC-DNN ([40])	100.00%	94.38%	71.80%	42.68%	34.85%	68.74%
SOM-SNN ([39])	99.60%	79.15%	36.25%	26.50%	19.55%	52.21%
This Work (Sine)	99.53%	99.38%	90.5%	47.68%	20.38%	71.49%
This Work (Linear)	99.73%	98.70%	77.85%	41.68%	18.18%	67.23%

**Table 5 brainsci-13-00319-t005:** Parameters set of Mnist and Fashionmnist dataset.

Network Parameter	Description	Value
$V_{th}$	Threshold of spiking neurons	1.0
epochs	Training epochs	200
batch size	The number of samples selected for a training	100
$T_{max}$	One quarter of the learning rate decay period	100
$lr_{ini}$	The initial value of learning rate	5 × 10${}^{-2}$
$lr_{min}$	The minimum value of learning rate	1 × 10${}^{-4}$

**Table 6 brainsci-13-00319-t006:** Network performance under different encoding numbers.

Time Window	Accuracy	Time Window	Accuracy
8	99.32%	14	99.60%
10	99.35%	16	99.46%
12	99.37%	18	99.51%

**Table 7 brainsci-13-00319-t007:** Comparison with similar-architecture SNNs on MNIST dataset.

Model	Methods	Network Structure	Time Window	Accuracy
STBP [17]	Spike-based BP	15C5-P2-40C5-P2-FC300-FC10	30	99.42%
HM2-BP [44]	Spike-based BP	15C5-P2-40C5-P2-FC300-FC10	400	99.49%
ST-RSBP [45]	Spike-based BP	15C5-P2-40C5-P2-FC300-FC10	400	99.62%
Lee et al. [32]	Spike-based BP	20C5-P2-50C5-P2-FC200-FC10	120	99.59%
ASF-BP [46]	Spike-based BP	20C5-P2-50C5-P2-FC200-FC10	300	99.65%
Relaxation LIF [47]	Spike-based BP	15C5-P2-40C5-P2-FC300-FC10	10	99.53%
This work (without SDE)	Spike-based BP	15C5-P2-40C5-P2-FC300-FC10	14	99.52%
This work (with SDE)	Spike-based BP	15C5-P2-40C5-P2-FC300-FC10	14	99.60%

**Table 8 brainsci-13-00319-t008:** Comparison with other SNNs model over Fashion-MNIST dataset.

Model	Network Structure	Time Window	Accuracy
ST-RSBP [45]	400-R400	400	90.13%
GLSNN [48]	256×8	10	89.02%
LISNN [49]	32C3-P2-32C3-P2-FC128-FC10	20	92.07%
STiDi-BP [50]	20C5-P2-40C5-P2-1000-10	100	92.80%
This work	15C5-P2-40C5-P2-FC300-FC10	20	90.26%
This work	15C5-P2-40C5-P2-FC300-FC10	100	91.71%

**Table 9 brainsci-13-00319-t009:** The performance of the proposed SNNs under different mapping methods with different training set proportions.

Methods	Time Window	Ratio	Clean	20 dB	10 dB	0 dB	−5 dB	Average
Linear	14	0.9	99.70%	98.88%	80.38%	40.88%	20.00%	68.00%
Linear	14	0.1	98.97%	97.35%	76.07%	43.55%	18.75%	66.90%
Sine	12	0.9	99.75%	99.625%	92.125%	50.50%	24.00%	73.20%
Sine	12	0.1	97.292%	97.318%	88.916%	47.21%	21.915%	70.53%

## Data Availability

The dataset we used in this study is the RWCP, MNIST and Fashion MNIST dataset, and they are openly available in http://research.nii.ac.jp/src/en/RWCP-SSD.html, http://yann.lecun.com/exdb/mnist and https://github.com/zalandoresearch/fashion-mnist (accessed on 10 November 2022).

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
