# Peer review of "Constrain Bias Addition to Train Low-Latency Spiking Neural Networks"

_brainsci, 2023, doi:10.3390/brainsci13020319_

Round 1

Reviewer 1 Report

In the manuscript by Lin et al., to train backpropagation in spike neural networks, the authors proposed a new spike encoding method and derived a new learning algorithm by constraining the addition of bias voltage in only spiking neurons. The proposed methods were tested in different data sets and achieved promising results. Overall, this manuscript is satisfactory. Here are my suggestions to improve it.

Major Comment

In the “training with less data” section, the authors used different numbers of training data for the sound dataset but adjusted the ratios of training data to test data for MNIST and Fashion-MNIST data sets. The authors need to justify why they used different strategies. For the sound data set, the spectrum (or features) is less overlapping between different types of sound. It can be predicted that it does not need much training data to achieve satisfactory accuracy. The classification accuracy for the MINST and Fashion-MNIST data set should need a relatively large amount of training data given their more likely overlapping spectrum (or features). I believe the ratio of training data to test data may not matter when the training data size (the number) is large enough.

Minor comments

1, line 22, it is unclear what the authors mean by “the generation of the spike sequence is an asynchronous process.”

2, near lines 32-33, information encoding in biological neural networks is more complicated than the authors described. See the relevant work showing multiplexed coding by Naud & Sprekeler, PNAS 2018 in pyramidal neurons; Zang et al., eLife 2020, Zang & De Schutter, J. Neurosci 2021 in Purkinje neurons.

3, it is unclear what the authors mean by “its value is negatively correlated with the encoding time” in line 34. Does it mean the spiking latency?

4, lines 35-36, please clarify that the limited capacity is only specific to rate coding instead of time coding.

5, line 59, the comment regarding Tempotron is unfair. The network can definitely be extended to more layers, in principle.

6, in section 2.1, “can realize the accurate simulation of the SNNs,” this is inaccurate. LIF models can only approximate certain neuronal properties.

7, in lines 95-97, the new reset mechanism is not only more appropriate for training deep SNNs but also makes more biological sense. See the voltage-dependent afterhyperpolarization in Zang et al., Cell Rep 2018.

8, line 151, “is very unreasonable” is not correct. This is an undesired property in this neural network, but it does not mean it is unreasonable. Many neurons generate spontaneous spikes.

9, provide units for Table 1.

10, the authors are encouraged to share their model code.

Author Response

Major Comment

Q:  In the “training with less data” section, the authors used different numbers of training data for the sound dataset but adjusted the ratios of training data to test data for MNIST and Fashion-MNIST data sets. The authors need to justify why they used different strategies. For the sound data set, the spectrum (or features) is less overlapping between different types of sound. It can be predicted that it does not need much training data to achieve satisfactory accuracy. The classification accuracy for the MINST and Fashion-MNIST data set should need a relatively large amount of training data given their more likely overlapping spectrum (or features). I believe the ratio of training data to test data may not matter when the training data size (the number) is large enough.

Reply:

In this chapter, we adjusted the ratio of training set to test set of RWCP data set to 0.9:0.1 and 0.1:0.9 respectively, and adjusted the ratio of MNIST and Fashion-MNIST data set to 0.1 and 0.5 respectively.

Due to the small number of samples in the RWCP dataset (800 samples in total between the training set and the test), we believe that it is necessary to reduce the number of training samples to further check the effect of the network. For the ten categories of data we selected, on the one hand, it is convenient to compare the effect with other SNNS using this data set; on the other hand, the total number of samples for the ten categories of data is close to 1000, and each category is close to 100. In each experiment, we randomly selected 80 samples for each category, totaling 800 samples. Since both the training set and the test set came from these 800 samples, we believed that multiple trials and adjusting the ratio of the training set to the test set could serve as a cross-validation effect, and our results were the average of the ten trials. In our experiment, the proportion of the training set in the total sample was traversed from 0.1 to 0.9. Finally, we report experimental results with ratios of 0.1 and 0.9 in the paper.

For MNIST and Fashion-MNIST data sets, since the test set and training set are not mixed together, the number of test set samples will not be affected by adjusting the number of training set samples. We believe that reducing the number of training samples on these two data sets can still check the learning ability of the network, although it is not possible to achieve cross-validation like the RWCP data set. The reason why the selected proportion is inconsistent with RWCP data set is that when the proportion of training set samples is set at 0.9, the performance of the network is not significantly affected. Therefore, we further reduce the proportion of training samples and adjust it to 0.5, at which time the network can train normally. For MNIST and Fashion-MNIST, there is an obvious difference when the proportion of training samples is 0.1. MNIST is still trained normally, but there is an overfitting phenomenon in the Fashion-MNIST dataset. We think this is worth pointing out in the article, so we only report the web presence with a ratio of 0.5 to 0.1.

Minor comments

Q1: line 22, it is unclear what the authors mean by “the generation of the spike sequence is an asynchronous process.”

Reply: What we mean is, for the same layer of spike neurons, they don't fire at the same time. For temporal coding, different input data values will lead to different firing times of each spike neuron. For rate coding, it's hard to get all the neurons to do the same action at multiple time steps.

Q2: near lines 32-33, information encoding in biological neural networks is more complicated than the authors described. See the relevant work showing multiplexed coding by Naud & Sprekeler, PNAS 2018 in pyramidal neurons; Zang et al., eLife 2020, Zang & De Schutter, J. Neurosci 2021 in Purkinje neurons.

Reply:  Before introducing the encoding of SNN we add the following sentence and cite the following three articles: 

“In the field of neuroscience, neural information coding is concerned with the relationship between input signals and the response of individual or group neurons”

Naud, R.; Sprekeler, H. Sparse bursts optimize information transmission in a multiplexed neural code. Proceedings of the National Academy of Sciences 2018, 115, E6329–E6338

Zang, Y.; Hong, S.; De Schutter, E. Firing rate-dependent phase responses of Purkinje cells support transient oscillations. Elife 2020, 9, e60692.

Zang, Y.; De Schutter, E. The cellular electrophysiological properties underlying multiplexed coding in Purkinje cells. Journal of Neuroscience 2021, 41, 1850–1863.

Q3: it is unclear what the authors mean by “its value is negatively correlated with the encoding time” in line 34. Does it mean the spiking latency?

Reply:  What we mean is that the higher the value of the input data, the earlier the spike is emitted.

Q4: lines 35-36, please clarify that the limited capacity is only specific to rate coding instead of time coding.

Reply: We have removed the reference to the limited capability of timing encoding and added the following sentence to the section describing rate encoding:

"Rate coding generally requires a sufficient number of time steps to improve coding accuracy, which increases network power consumption."

Q5: line 59, the comment regarding Tempotron is unfair. The network can definitely be extended to more layers, in principle.

Reply: We modify the introduction about Tempotron algorithm to "Tempotron is a gradient descent learning algorithm based on biological characteristics that is suitable for temporal coding, when a classification error occurs, it updates the synaptic weight with temporal information. "

Q6: in section 2.1, “can realize the accurate simulation of the SNNs,” this is inaccurate. LIF models can only approximate certain neuronal properties.

Reply: We change the description of LIF neurons to the following sentence:

"As the most commonly used spiking neuron, the Leaky Integrate-and-Fire neuron (LIF) adopts event-driven simulation strategy and has limited neural computing characteristics."

Q7: in lines 95-97, the new reset mechanism is not only more appropriate for training deep SNNs but also makes more biological sense. See the voltage-dependent afterhyperpolarization in Zang et al., Cell Rep 2018.

Reply: We add the following description:

"The subtraction-based mechanism is better suited to training deep SNNs and makes more biological sense."

and cite the article:

Zang, Y.; Dieudonné, S.; De Schutter, E. Voltage-and branch-specific climbing fiber responses in Purkinje cells. Cell reports 2018, 24, 1536–1549.

Q8: line 151, “is very unreasonable” is not correct. This is an undesired property in this neural network, but it does not mean it is unreasonable. Many neurons generate spontaneous spikes.

Reply: We modify the description as follows:

“In case 1, since the bias is permanently forced to zero, it is equivalent to ignoring the regulation effect of bias on the threshold; while in case 2, adding bias voltage at every time-step undoubtedly violates the event-driven characteristics of SNNs because When time T → ∞, in the absence of any input, the spiking neuron will spontaneously emit spikes, which has a negative impact on training and reasoning.”

Q9: provide units for Table 1.

Reply: We add section 3.2.1 to introduce the CQT algorithm and hyperparameter settings.

Q10: the authors are encouraged to share their model code.

Reply:  We will make it public as soon as possible.

Your comments have effectively improved the quality of our manuscripts, and we would like to express our sincere thanks to you.

Reviewer 2 Report

This paper introduces a novel encoding method with a spike-based learning rule. One of the advantages of this method is that it achieves good results with smaller time steps. 

I find the introduction well written, and I was pleased to learn more about spiking neural networks in this paper.

My concern goes mainly to the results section:

1. The structure of this section is unclear. It contains in fact the description of the data and methods for the experimental validation AND the results of the experiment. It would be more clear for the reader if the plan reflects this distinction. I recommend also adding an introduction paragraph to this section.

2. Complete the description of the datasets by adding the number of samples in each class. 

3.  Please provide a short explanation about Constant-Q transform (CQT) in 3.3.1

4. Please, discuss and justify the value of your hyperparameters. 

5. Could you justify the choice of your metric (accuracy, precision) or add other metrics to describe the performance of your method?

6. Could you also report the training/test time of the different methods?

Author Response

Q1. The structure of this section is unclear. It contains in fact the description of the data and methods for the experimental validation AND the results of the experiment. It would be more clear for the reader if the plan reflects this distinction. I recommend also adding an introduction paragraph to this section.

Reply: We add section 3.1.3 to introduce our experimental section.

Q2. Complete the description of the datasets by adding the number of samples in each class.

Reply: We explain the datasets in more detail:

“The MNIST dataset is one of the most classic datasets in deep learning, which contains ten different handwritten digits and the input size is 28 × 28. It contains ten classes, and the training set has a total of 60000 samples, each class contains 6000 samples. The test set contains 10000 samples, 1000 samples per class. The Fashion-MNIST and MNIST dataset have the same number of sample classes and samples. Its data is more complex, in which the content of the pictures includes various clothes, the input size is 28 × 28, and the number of training and test sets is the same as in MNIST.”

Q3. Please provide a short explanation about Constant-Q transform (CQT) in 3.3.1.

Reply: We add section 3.2.1 to introduce the CQT algorithm.

Q4. Please, discuss and justify the value of your hyperparameters. 

Reply: For the environmental sound classification experiment using the RWCP dataset, we set two hyperparameters, N_bin and T_seg, where N_bin refers to the number of frequency bins and T_seg refers to the final obtained  data per sub band. The setting of N_bin is mainly determined by B (described in Section 3.3.1). If we set B to a larger value, the final value of N_bin will also be large, and the size of the obtained spectrum diagram will also be large. To limit the size of the spectrogram, we set B to 3, so as to ensure that the frequency box with the maximum bandwidth obtained can cover a part of the region above 10Khz. At the same time, Sufficient frequency bins can also be obtained in the 1khz region.

Similarly, for the spectrogram just obtained through CQT, the sampling points of each sub band reach hundreds (in direct proportion to the length of audio). Therefore, to reduce the size of the final feature graph, we divide it into T_seg overlapping regions, and select the average value of each region to finally obtain the feature graph with the size of (N_bin ,T_seg). After flattening, Input the fully connected network, the number of neurons in the first layer of the fully connected network is N_bin * T_seg.

Q5.  Could you justify the choice of your metric (accuracy, precision) or add other metrics to describe the performance of your method?

Reply: We add 3.5 sections to analyse the four aspects of the network in terms of the running time and memory occupation of a single epoch, the performance of the network and the convergence speed, and compare it with the original IF neuron, highlighting the advantages of our model.

Q6.  Could you also report the training/test time of the different methods?

Reply: The time windows used by other SNN models in the MNIST and Fashion-MNIST data sets are listed in the table 7 and table 8. For the RWCP dataset, the rest of the SNN models use sequential encoding, so there is no time window.

We sincerely thank you for your affirmation of our work. Your comments have improved the quality of our manuscripts. Thank you very much for your help to us.

Reviewer 3 Report

1. the percentage of the improvement should be added at the end of the abstract.

2. The authors have to add efficiency analysis in terms of consumed time , memory, complexity,etc.

3. Samples of used datasets should be added.

4. improve quality of figures.

5. Add pseduo code of the whole approach.

6. Add clear and sound diagram of the proposed architecture.

Author Response

Q1. the percentage of the improvement should be added at the end of the abstract.

Reply: We added our improvement on the MNIST dataset at the end of the abstract.

Q2. The authors have to add efficiency analysis in terms of consumed time, memory, complexity,etc.

Reply: We added 3.5 sections to analyse the four aspects of the network in terms of the running time and memory occupation of a single epoch, the performance of the network and the convergence speed, and compared it with the primitive IF neurons, highlighting the advantages of our model.

Q3. Samples of used datasets should be added.

Reply: We explained the datasets in more detail:

“The MNIST dataset is one of the most classic datasets in deep learning, which contains ten different handwritten digits and the input size is 28 × 28. It contains ten classes, and the training set has a total of 60000 samples, each class contains 6000 samples. The test set contains 10000 samples, 1000 samples per class. The Fashion-MNIST and MNIST dataset have the same number of sample classes and samples. Its data is more complex, in which the content of the pictures includes various clothes, the input size is 28 × 28, and the number of training and test sets is the same as in MNIST.”

Q4. improve quality of figures.

Reply: We resized the image and redrew Figure 2. All images have a resolution of 300dpi.

Q5. Add pseudo code of the whole approach.

Reply: We added pseudo code at the end of Section 2.3.2.

Q6. Add clear and sound diagram of the proposed architecture.

Reply: We modified Figure 4 and raised its dpi to 600.

We would like to thank you for your comments on our work, which enhance the quality of our manuscripts and make it more rigorous.

Round 2

Reviewer 1 Report

The authors have solved all my concerns.

Reviewer 2 Report

The authors have addressed my previous comments.